# The Role of AI in Breast Cancer Lymph Node Classification: A Comprehensive Review

**DOI:** 10.3390/cancers15082400

**Published:** 2023-04-21

**Authors:** Josip Vrdoljak, Ante Krešo, Marko Kumrić, Dinko Martinović, Ivan Cvitković, Marko Grahovac, Josip Vickov, Josipa Bukić, Joško Božic

**Affiliations:** 1Department of Pathophysiology, University of Split School of Medicine, 21000 Split, Croatia; josip.vrdoljak@mefst.hr (J.V.); marko.kumric@mefst.hr (M.K.);; 2Department of Surgery, University Hospital of Split, 21000 Split, Croatia; ante.kreso@mefst.hr (A.K.); dinko.martinovic@mefst.hr (D.M.);; 3Department of Pharmacology, University of Split School of Medicine, 21000 Split, Croatia; 4Department of Pharmacy, University of Split School of Medicine, 21000 Split, Croatia

**Keywords:** breast cancer, lymph node classification, AI, artificial intelligence, machine learning, deep learning, radiomics, clinicopathological features

## Abstract

**Simple Summary:**

Breast cancer affects countless women worldwide, and detecting the spread of cancer to the lymph nodes is crucial for determining the best course of treatment. Traditional diagnostic methods have their drawbacks, but artificial intelligence techniques, such as machine learning and deep learning, offer the potential for more accurate and efficient detection. Researchers have developed cutting-edge deep learning models to classify breast cancer lymph node metastasis from medical images, with promising results. Combining radiological data and patient information can further improve the accuracy of these models. This review gathers information on the latest AI models for detecting breast cancer lymph node metastasis, discusses the best ways to validate them, and addresses potential challenges and limitations. Ultimately, these AI models could significantly improve cancer care, particularly in areas with limited medical resources.

**Abstract:**

Breast cancer is a significant health issue affecting women worldwide, and accurately detecting lymph node metastasis is critical in determining treatment and prognosis. While traditional diagnostic methods have limitations and complications, artificial intelligence (AI) techniques such as machine learning (ML) and deep learning (DL) offer promising solutions for improving and supplementing diagnostic procedures. Current research has explored state-of-the-art DL models for breast cancer lymph node classification from radiological images, achieving high performances (AUC: 0.71–0.99). AI models trained on clinicopathological features also show promise in predicting metastasis status (AUC: 0.74–0.77), whereas multimodal (radiomics + clinicopathological features) models combine the best from both approaches and also achieve good results (AUC: 0.82–0.94). Once properly validated, such models could greatly improve cancer care, especially in areas with limited medical resources. This comprehensive review aims to compile knowledge about state-of-the-art AI models used for breast cancer lymph node metastasis detection, discusses proper validation techniques and potential pitfalls and limitations, and presents future directions and best practices to achieve high usability in real-world clinical settings.

## 1. Introduction

Breast cancer is the most common cancer among women and a leading cause of cancer deaths among women worldwide [1]. Accurately determining the presence or absence of lymph node metastasis is crucial in determining the treatment and prognosis of breast cancer. Current diagnostic methods for determining breast cancer lymph node metastasis include sentinel lymph node dissection (SNLD), axillary lymph node dissection (ALND), and radiological (ultrasound, MRI, CT) and nuclear medicine techniques [2]. SLND is the gold standard for predicting axillary lymph node (ALN) status, and it is a recommended technique, particularly in patients with clinically negative nodes [3].

However, each aforementioned diagnostic method has certain limitations and complications. For example, radiological methods misdiagnose up to 30% of patients, and ALND is a large surgical procedure and can lead to lymphedema, loss of sensation in the arm, and movement restriction. Furthermore, the ACOSOG Z0011 trial has shown that in patients with T1/T2 breast cancer who have ≤2 SLN metastases, the use of SLND without ALND would not lead to inferior survival [4]. Nevertheless, while SLND is less invasive, it can still lead to arm numbness or upper limb edema, together with the accompanying costs and anesthesia risks. Hence, supplementary diagnostic methods are still needed to improve our diagnostic procedures and, concurrently, lead to better treatment outcomes.

Artificial intelligence, and more specifically machine learning (ML) and deep learning (DL), with their current advancements, are prime candidates for improving and supplementing the traditional diagnostic methods for ALN metastasis detection [5,6,7]. A lot of new research has explored the development of state-of-the-art DL models for breast cancer detection from radiological images. For example, deep convolutional neural networks (CNN) combined with fuzzy ensemble modelling techniques have achieved an accuracy rate of 99.32% in detecting breast cancer from mammograms [8]. Moreover, AI techniques have also been applied to clinicopathological features (e.g., patient age, tumor size, ER, PR, HER-2) with the goal of predicting breast cancer lymph node metastasis status. When AI models were trained only on clinicopathological features, area-under-the-curve (AUC) scores reached up to 0.80, whereas when clinicopathological features were combined with radiological methods, performances were significantly increased, with an AUC of 0.90 [9,10]. Given the rapid pace of growth in the ML/DL field, we can expect model accuracies to keep improving. Hence, such models could help us achieve better diagnostics, as well as better treatment outcomes, especially in areas/countries that have limited medical resources. For instance, tree-based models such as random forests and eXtremeGradientBoosting (XGBoost) can be trained and deployed on relatively cheap computer hardware, without the need for expensive graphical processing units (GPUs) and computer servers [11]. Therefore, once adequately validated, such models could greatly improve cancer care in a limited-resource setting.

In this comprehensive review, our goal is to compile knowledge about state-of-the-art AI models used for breast cancer lymph node metastasis detection. We will discuss modeling and proper validation techniques, and review the potential pitfalls and limitations of such models. Furthermore, we will present future directions and best practices to overcome such limitations, and hopefully reach high usability in a real-world clinical setting.

## 2. Search Methodology

This study is a comprehensive review of the published literature focusing on ML/DL in breast cancer lymph node classification. We searched the following databases: PubMed, Google Scholar, Web of Science, and Scopus. We included studies written in English, and the year of publication restriction was not applied. The keywords used in the search were “Breast cancer”, “Lymph node prediction”, Lymph node classification”, “Machine learning”, “Deep learning”, and “Artificial intelligence or AI”. After performing the search, 52 studies were selected from initial 431 studies for further examination. For additional studies, we examined reference lists of relevant research papers.

## 3. ML Methods Used for Breast Cancer Lymph Node Classification

### 3.1. Approaches Used in Radiomics

#### 3.1.1. Transfer Learning

Similar to applications that use computer vision for breast cancer detection, the most common way new models for lymph node detection are developed is by utilizing transfer learning [12]. Transfer learning is a particular subset of machine/deep learning in which the intricacies (“weights”) learned by a large model are transferred to a new model, which is then fine-tuned for the specific desired task [13]. For example, large CNNs such as VGG16, VGG19, and ResNet are trained on a very large pool of image data. The usual benchmark to test these networks is the ImageNet dataset, which contains 1000 different classes (from animals to people and other everyday objects) [14]. It has been shown that the base knowledge about vision learned from these networks, can be efficiently transferred and reused by a new network. It is argued that the early layers of such neural networks learn the basics, such as recognizing edges and common shapes. Therefore, in transfer learning, we usually leave the majority of the large model’s architecture intact, while we discard the last or the last few neural network (NN) layers and redesign/retrain them for our purposes.

Such an approach has proven to be effective, and much more computationally and resource efficient than training a new neural network from scratch [15], especially given the relative scarcity of medical image data.

#### 3.1.2. Training a Convolutional Neural Network De Novo

Another approach researchers can take is to train a new CNN just on their dataset. Such an approach is good if the researchers can access a large pool of training data, which, as mentioned before, is often not the case in medical imaging. Studies that trained a new CNN in the setting of breast cancer lymph node detection from radiological images often had from 100 to 1000–2000 training images [16,17,18], whereas ResNet was trained on more than one million images [19]. On the other hand, when training a de novo network, researchers can experiment with model architectures and potentially develop architectures that are better optimized for the task at hand.

#### 3.1.3. Training Traditional ML Algorithms

The final approach that is used in radiomics is to train a non-deep learning algorithm, such as logistic regression, tree-based models (random forest, XGBoost), or support vector machines (SVM) [20,21]. This approach was common in older studies, but has now given way to training CNNs and transfer learning, which are usually much better performers at computer vision tasks [22]. Additionally, when training a traditional ML algorithm for such tasks, features often need to be manually extracted and preprocessed, whereas DL algorithms handle feature extraction automatically [22].

### 3.2. Approaches Used When Training Only on Clinicopathological Features

Training models for lymph node detection based only on clinicopathological features means that input (training) data are in tabular form. The usual clinicopathological features in breast cancer are tumor size, patient age, the status of estrogen, progesterone and epidermal growth factor receptors (ER, PR, HER-2), tumor grade, histological type and immunophenotype.

When dealing with tabular data, ensemble tree-based models such as random forests and gradient boosted trees (XGBoost, CatBoost) have outperformed DL models (neural networks) [23,24]. Therefore, it is usually better to use tree-based models when training only on clinicopathological data. Nevertheless, neural networks, with optimized architectures and hyperparameters, can also achieve good performance in breast cancer lymph node detection, as was shown in a paper by Dihge et al., (AUC: 0.705–0.747) [25].

### 3.3. Best Practices for Model Validation

When trying to ascertain how a model would perform in a real-world setting on previously unseen data, proper validation techniques must be used. Through such validation techniques, dispersion metrics such as 95% confidence intervals and standard deviations are obtained. Hence, a simple train/test split is not enough, and can often lead to overfitting the data and obtaining overly optimistic results.

The first approach researchers can take to properly validate their models is to perform repeated cross validation [26]. Cross validation (CV) is a technique in which the dataset is divided into N folds, and in each fold the data are split into training and test sets at different non-overlapping indexes. For example, we can use five-fold cross-validation, in which five folds are created, and in each of the five folds a different 20% of the data is used for the test set.

After repeating the CV N times, we can obtain dispersion metrics and valuable estimates about the model’s true performance.

Another approach by which dispersion metrics can be ascertained is by performing bootstrapping. Bootstrap is a general technique for estimating statistics that can be used to calculate confidence intervals, regardless of the underlying data distribution. In bootstrapping, a new dataset is obtained by performing resampling with the replacement of the original training or test data. Therefore, we obtain N (for example 1000) new datasets to train or evaluate the models on, after which dispersion metrics can be calculated [27].

Confidence intervals can also be calculated when assuming a normal data distribution, using the traditional formula sample mean±z∗standard error. Since the standard error (SE) is inversely proportional with the sample size, we often obtain a small SE in dataset sizes usually implemented in ML/DL, and therefore narrower confidence intervals. Furthermore, such a calculation is performed only on one test set, and hence is also subject to potential bias [26].

CV or bootstrapping are mandatory in order to properly validate a ML model, and should be present in every study.

In the next part, we will summarize the results and limitations of the latest state-of-the-art studies concerning breast cancer lymph node prediction with ML and DL.

## 4. Studies Using Radiomics for Breast Cancer Lymph Node Prediction

The most common radiological methods used in radiomics for classifying axillary lymph nodes are MRI and ultrasound. Firstly, we will summarize the studies that used ML/DL on MRI acquired data.

MRI-based radiomics approaches have achieved very good results in predicting axillary lymph node metastasis. A recent study by Wang et al. evaluated a model trained on T1-weighted imaging (T1WI), T2-weighted imaging (T2WI), and diffusion-weighted imaging (DWI) sequences [28]. The dataset used in the study contained 348 breast cancer patients, 163 with axillary lymph node metastasis, and 185 patients without metastasis. The data were randomly split into the training set (315 cases) and the testing set (33 cases). A Transfer learning approach was used, where the data were firstly applied to a pretrained ResNet50 model, and then the results of different sequences were sent to an ensemble learning classifier which generated the final results. The ensemble model achieved an AUC of 0.996 and an accuracy of 0.970 (Table 1) [28]. While the authors did perform a five-fold CV on the training set, they did not perform repeated outer CV to obtain dispersion metrics. A small sample size and a lack of proper validation could induce bias and produce overly optimistic results.

Another recent study, by Zhang et al., has also reported excellent results, with an AUC of 0.913 (95% CI: 0.799–0.974) (Table 1) [29]. A total of 252 patients were randomly divided into a training (202) and test set (50), and Transfer learning (with pre-trained ResNet50) was used. Finally, T2WI, DWI, and DCE-MRI sequences were used, and their results were pooled to perform the final classification [29]. While 95% confidence intervals were reported, it was not specified whether CV or bootstrapping was performed. Again, overly optimistic results could be obtained due to a small sample size and a lack of a validation cohort or CV/bootstrapping.

An interesting study by Gao et al. investigated the potential of a new attention-based DL model [30]. The model combined Transfer learning (ResNet50) with a convolutional block attention module (named RCNet). The dataset consisted of 941 breast cancer patients who underwent DCE-MRI before surgery (742 training set, 83 internal test set, 116 external test set) [30]. The model achieved an AUC of 0.852 (0.779–0.925) when evaluated on the external test set (Table 1). Notably, the authors also showed how the model improves radiologist performance in a statistically significant manner [30].

Another study also showed favorable results when comparing a DL CNN model’s performance with that of an expert radiologist [16]. In the study, Ren et al. trained a CNN on standard clinical breast MR images, with a goal to detect nodal metastasis [16]. Data used in the study consisted of 66 abnormal nodes from 38 patients and 193 normal nodes from 61 patients. Node abnormality was determined by an expert radiologist based on 18-Fluorodeoxyglucose positron emission tomography images. The model achieved an AUC of 0.91 ± 0.02 on the validation set, and an accuracy of 84.8% ± 2.4%, whereas the radiologist achieved an accuracy of 78% when examining the same MRI images (Table 1) [16]. While the model did show an increase in accuracy compared to a radiologist, this is not the correct metric to use in this case due to class imbalances (66 abnormal nodes vs. 193 normal nodes) [31].

In a later study, the same authors evaluated a CNN trained on multiparametric MRI in breast cancer patients, namely using T1-W MRI, T2-W MRI, DCE MRI, T1-W + T2-W MRI, and DCE + T2-W sequences [17]. The best performances were achieved by a model trained on T1-W + T2-W MRI (accuracy = 88.50%, AUC = 0.882) (Table 1). All models achieved better accuracy than a radiologist (accuracy = 65.8%). Nevertheless, similarly to the previous study, the sample size was small (just 56 patients, with 238 axillary lymph nodes), hence potentially providing biased results [17].

A similar approach with a CNN model trained on DCE MRI samples was used in a study by Santucci et al. [32]. The study was based on the concept of peritumoral tissue containing useful information about tumor aggressiveness. A total of 155 patients were examined, with 27 patients having positive lymph nodes and 128 patients having negative ones [32]. The authors evaluated different models by performing a 10-fold cross validation, and the best CNN achieved a test AUC of 0.789 (Table 1). Due to the small sample size, with only 27 positive cases, a 10-fold cross validation would lead to validation and test folds that contain only a few (or none) examples of a positive case. Hence, metrics that are sensitive to class imbalance, such as accuracy, will provide biased results. Furthermore, while the authors did report the mean results of the tested metrics, they did not report any measures of dispersion (e.g., 95% CI, SD). Therefore, we cannot estimate how the model would perform in a real-world setting.

Ha et al. also evaluated CNN performances in classifying metastatic lymph nodes from MR images [18]. Axillary lymph nodes were identified on the first T1 post contrast dynamic images. A total of 275 axillary lymph nodes were used for the study (133 LN +, 142 LN-). Five-fold cross validation was performed, and mean accuracy was reported (84.3%) (Table 1). The authors did not report the AUC and dispersion metrics [18].

One study investigated the performances of random forest models trained on features extracted from delineated lymph nodes for MRI-radiomics [33]. They tested 100 different models, and the final results in the test cohort ranged from 0.37 to 0.99 (Table 1). Due to a wide range in performances, it was not possible to obtain a final prediction model [33]. Hence, dedicated axillary MRI-based radiomics with node-by-node analysis were not efficient in breast cancer axillary lymph node classification.

**Table 1 cancers-15-02400-t001:** Summary of studies using MRI radiomics for breast cancer lymph node classification.

Study	Sample Size	Imaging Approach	Model †	Metric
Wang et al. [28]	348	T1WI, T2WI, DWI	ResNet50+ Ensemble	AUC = 0.996
Zhang et al. [29]	252	T2WI, DWI, DCE	ResNet50+ Ensemble	AUC = 0.913 (0.799–0.974)
Gao et al. [30]	941	DCE	ResNet50 + RCNEt	AUC = 0.852 (0.779–0.925)
Ren et al. [16]	259 nodes (from 99 patients)	Standard breast MRI	CNN de novo	AUC = 0.91 ± 0.02
Ren et al. [17]	238 nodes (from 56 patients)	T1WI, T2WI, DCE, T1 + T2, DCE + T2	CNN de novo	AUC = 0.882
Santucci et al. [32]	155	DCE	CNN de novo	AUC = 0.789
Ha et al. [18]	275 nodes (from 142 patients)	T1-post contrast dynamic	CNN de novo	Accuracy = 0.843
Samiei et al. [33]	511 nodes (from 75 patients)	T2WI	Random forests (100 models)	AUC = 0.37–0.99

AUC—area under the ROC curve, CNN—convolutional neural network, MRI—magnetic resonance imaging, T1WI—T1-weighted imaging, T2WI—T2-weighted imaging, DWI—diffusion-weighted imaging, DCE—Dynamic contrast-enhanced. † All studies used Python programming language for modelling purposes (along with “Keras”, “TensorFlow” and “PyTorch” libraries).

In the following part, we will summarize the studies that utilized AI-approaches on ultrasound images to detect axillary lymph node metastasis.

A study by Lee et al., utilized a deep-learning-based, computer-aided prediction system for ultrasound (US) images [34]. A total of 153 women with breast cancer were involved in the study (59 patients LN+,94 patients LN−). The authors trained and evaluated several algorithms (logistic regression, SVM, XGBoost, DenseNet) on US image data, and the best AUC was achieved with DenseNet (0.8054) (Table 2) [34]. Cross-validation or bootstrapping was not performed and dispersion metrics were not reported, and hence the true validity of the results remains unknown.

An interesting study by Ozaki et al. evaluated the performances of a DL-model using a CNN Xception architecture [35]. The model scored an AUC of 0.966, which was comparable with to the results scored by a radiologist with 12 years of experience (0.969; *p* = 0.881) [35]. On the other hand, the model scored better than two resident radiologists with 3 years and 1 year of experience (0.913; *p* = 0.101 and 0.810; *p* < 0.001) (Table 2.), thus indicating the potential for useful diagnostic support for radiology residents [35].

Somewhat worse performances were achieved in a study by Sun et al., where a CNN was trained and tested on ultrasound images from 169 patients [36]. A total of 248 US images from 124 patients were used in the training dataset, whereas 90 US images from 45 patients were used in the test dataset. The authors reported an AUC of 0.72 (SD ± 0.08) and an accuracy of 72.6% (SD ± 8.4) (Table 2) [36]. Again, cross validation and bootstrapping were not used in the validation process.

A comparison between CNNs and traditional ML methods (random forests) was made in another study performed on 479 breast cancer patients with 2395 breast ultrasound images [37]. Moreover, the study focused on different parts of US images, where intratumoral, peritumoral and combined regions were utilized to train and evaluate the models. CNNs performed better than random forests in all modalities (*p* < 0.05), and combining intratumoral and peritumoral regions yielded the best result (AUC = 0.912 [0.834–99.0]) (Table 2) [37]. Confidence intervals were reported, but it was not stated by which method they were obtained.

One study compared the performances of Google Cloud AutoML Vision (Mountain View, CA, USA) with the performances of three experienced radiologists [38]. Ultrasound images of 317 axillary lymph nodes from breast cancer patients were collected. When evaluated on three independent test sets, the AI system achieved better specificity (64.4% vs. 50.1%) and positive predictive value (68.3% vs. 65.4%), while it achieved worse sensitivity (74.0% vs. 89.9%) and comparable accuracy (69.5% vs. 70.1%) (Table 2) [38]. Hence, this study showed how an automated AI system yields performances similar to that of experienced radiologists.

A recent study by Zhang et al. evaluated the performances of 10 different ML algorithms trained on ultrasonographic features of 952 breast cancer patients [39]. The primary cohort was randomly divided into 10 parts, and 10-fold cross validation was performed to calculate metrics such as average AUC and average accuracy [39]. XGBoost was the best-performing algorithm, with an average validation AUC of 0.916 (Table 2). Nevertheless, repeated cross validation was not performed and dispersion metrics were not derived, thus limiting the applicability of the results. On the other hand, this study used a comparatively larger sample size than previously mentioned ultrasound-radiomics studies. Moreover, the authors used an interesting framework for model explanation that stems from game theory: SHapley Additive exPlanation (SHAP). Shapley values identified a suspicious lymph node as the most important ultrasound feature, followed by margin, architectural distortion, and calcification [39].

Deep learning radiomics of ultrasonography model (DLRU) also showed great promise in stratifying patients into low-risk and high-risk categories for axillary metastasis, as is shown in a study by Guo et al., where a DL-model was trained and validated on 937 eligible breast cancer patients with ultrasound images [40]. The model yielded high performances in identifying sentinel lymph node metastasis (SNL) (sensitivity = 98.4% [96.6–100]), and in non-sentinel metastasis (sensitivity = 98.4% [95.6–99.9]) [40]. Moreover, the model accurately stratified patients into low-risk (LR)-SLN and high-risk (HR)-SLN&LR-NSLN categories, with the negative predictive value of 97% [94.2–100] and 91.7% [88.8–97.9], respectively (Table 2) [40]. This accurately identified overtreated patients and provided utility to avoid overtreatment. The authors further argue that given the relative simplicity and lower cost of ultrasound imaging, such methods could be helpful for hospitals in limited-resource settings [40].

Aforementioned studies exhibit how ML/DL approaches combined with ultrasound radiomics provide very good performances (AUC 0.72–0.97) and have high potential for clinical utility. Nevertheless, relatively small sample sizes and a common lack of proper validation techniques give way to biased results. Hence, prospective validation and new multicentric studies on larger sample sizes are needed to ascertain the usefulness of ultrasound radiomics in everyday clinical practice.

Now, we will briefly summarize the studies that used other radiological methods (CT, pet-CT, mammography) with ML/DL to diagnose metastatic lymph nodes in breast cancer patients. A proof of principle study was conducted on a small breast patient cohort (75 patients), where researchers investigated how deep convolutional neural network (dCNN) can accurately detect abnormal axillary lymph nodes on mammograms [41]. After training and validating the dCNN, it was tested on a “real-world” dataset for the presence of three different classes (breast tissue, benign lymph nodes and suspicious lymph nodes). It yielded an accuracy of 98.51% for breast tissue, 98.63% for benign lymph nodes, and 95.96% for suspicious lymph nodes (Table 3), where the ground truth was established through radiological reports. Hence, the limit of this model’s performance is the same as the limit of conventional radiological performances.

One study used DL on contrast-enhanced computed tomography (CECT) images (800 CECT images from 401 breast cancer patients) [42]. Transfer learning was utilized, and a deformable attention VGG19 (DA-VGG19) neural network was evaluated. The validated model achieved excellent accuracy of 0.908, while the AUC achieved 0.805 (Table 3.) [42]. For validation and hyperparameter optimization, the authors implemented a five-fold CV, and no dispersion metrics (SD and 95%CI) were reported.

Another study implemented deep learning techniques on contrast-enhanced CT preoperative images from 348 patients [43]. Relevant image features were extracted to establish the deep learning signature of SLN metastasis. Good performance was achieved on the validation cohort, with an AUC of 0.817 (95% CI: 0.751–0.884) (Table 3).

Furthermore, a radiomics model where XGBoost was trained on PET/CT images from 100 patients achieved relatively high sensitivity, specificity, and accuracy (90.9%, 71.4%, and 80%, respectively) [44]. The AUC was 0.890 (95%CI: 0.700–0.979) (Table 3), while it was not stated if CV or bootstrapping were performed [44]. The diagnostic performances of PET/CT were 55.8%, 93%, and 77%, for sensitivity, specificity, and accuracy, and hence the model exhibited better performances than radiologists.

Recently, a random forest model was trained and evaluated on image data gathered from PET/MRI (303 participants from 3 centers) [45]. The model achieved better but statistically not significant diagnostic accuracy when compared to radiologists (89.3% vs. 91.2%, *p* = 0.683) (Table 3) [45].

Many aforementioned studies utilize multiple images from individual patients; in these situations, it is crucial to prevent data leakage by dividing the training and validation/test sets based on subjects, ensuring that no single patient’s data appears in both the training and test sets.

Overall, model performances when trained on mammography, CT, or PET-CT/MRI data show promising results, and such models could be implemented in clinical workflow after further validation.

## 5. Studies Using Clinicopathological Features for Breast Cancer Lymph Node Prediction

The second approach that utilizes ML/DL techniques for breast cancer lymph node classification is training a model on only patient and tumor clinicopathological features. Those clinicopathological features are extracted during a regular pre-operative or pre-chemotherapy patient workup. Pathological features are obtained through tumor biopsy, tumor size is obtained through imaging techniques, and other patient features are obtained through a regular medical examination. Common features are patient age, ER, PR and HER-2 status, Ki-67 index, tumor size, histological type, immunophenotype, and the presence of lymphovascular invasion. Current research has shown that training a ML/DL model on only clinicopathological features can achieve good results (AUC 0.74–0.77) in lymph node classification [10,25,46] (Table 4). Specifically, a study by Dihge et al., which was performed on a total of 800 patients, developed a neural network that achieved an AUC of 0.74 [0.72–0.76] (Table 4). In their study, the two clinicopathological features with the highest predictive value were tumor size and the status of lymphovascular invasion (LVI).

On the other hand, in our previous study, ML models were trained on clinicopathological data without (LVI), gathered from 8381 patients [10]. Tree-based models such as random forest and XGBoost achieved the best performances (AUC 0.762 [0.726–0.795] and 0.760 [0.724–0.794], respectively) (Table 4), and feature importance was determined with Shapley values, where tumor size, Ki-67, and patient age were the features with the highest predictive power [10].

One other study that evaluated ML methods only on breast cancer patients with invasive micropapillary carcinoma also had XGBoost as the best performing algorithm [47]. As determined by Shapley values, tumor size, and patient age were the two most important features. Models were trained on data extracted from 1405 patients and evaluated on an external validation cohort of 142 patients. Interestingly, while XGBoost achieved an AUC of 0.76 [0.746–0.776] on the training set and an AUC of 0.81 [0.799–0.826] on the test set, when evaluated on an external validation cohort the AUC dropped to 0.700 [0.683–0.716], hence indicating that the model was overfitted on the previously seen data [47].

Moreover, an older study by Takada et al. also utilized a tree-based model (ADTree) trained on clinicopathological features, and achieved an AUC of 0.77 [0.69–0.86]. Since clinicopathological features for breast cancer lymph node prediction are presented in tabular form, it is expected that random forest (RF) and gradient-boosted trees (GBT) models will generally outperform other model architectures [23,24]. This is especially true for small- to middle-sized datasets (~10,000 samples), where RF and GBT remain robust to uninformative features, whereas NNs are sensitive to such features and hence generalize poorly on unseen data [24]. Therefore, when dealing with clinicopathological data, a general approach would be to use tree-based models.

While the current performance of models trained on clinicopathological features underperforms the radiomics models, the addition of new genetic/epigenetic features and novel tumor markers could improve performance. Even simple features such as weight and BMI are underused in this setting, and they could prove beneficial because research has shown a connection between greater BMI and a higher chance of lymph node metastasis [48]. Moreover, because the features used to train such models are generally obtained during regular breast cancer workup, they do not add additional operational costs.

## 6. Studies Combining Radiomics and Clinicopathological Features

The last ML/DL approach for breast cancer lymph node prediction combines radiomics and clinicopathological approaches. Hence, a model is trained on radiomic features extracted from radiologic images (for example MRI or US), and on clinicopathological features.

Such an approach was utilized in a study by Zheng et al., where a DL model was trained on features extracted from US images, combined with clinicopathological features [9]. The model trained on both feature modalities achieved statistically higher performance than the models trained only on US image data or only on clinicopathological data (AUC 0.902 vs. 0.735 and 0.796, respectively) (Table 5) [9].

Another study also exhibited excellent model performance when trained on a combination of PET-CT and clinical data [49]. The model was trained on data from 203 patients, while data from 87 patients were held out for the validation set. The integrated model performed with an AUC of 0.93 (95% CI, 0.88–0.99) on the validation set (Table 5). The authors performed 10-fold cross-validation to select optimal features for predicting ALN metastasis, which was an overly high number of CV-folds given the small sample size (290 patients) [49].

An interesting study by Chen et al. combined DCE-MRI radiomics data with transcriptomic data from The Cancer Genome Atlas for a set of 111 patients with breast cancer [50]. Another fifteen patients were enrolled for the external validation group. When evaluated in the validation group, the radio-genomics model showed increased performances compared to both the radiomics-only and genomics-only models (AUC 0.82 vs. 0.71 and 0.52, respectively) (Table 5). The authors used logistic regression as the model of choice and performed a five-fold inner CV to optimize the model hyperparameters, whereas no dispersion metrics were reported for the results [50]. It remains to be seen whether a model with a more complex architecture would better capture the intricacies of multi-modal data.

Generally, combining image data (radiomics) and clinicopathological data yields better and more robust results. Accordingly, building multi-modal models that utilize multiple data sources (such as image, clinicopathological, and genetic data) will presumably lead to more accurate predictive models that will outperform current gold standard methods. On the other hand, utilizing such models is a resource intensive endeavor which will probably only be feasible in the largest clinical centers.

## 7. Discussion and Future Perspectives

We are witnessing exponential growth in methods that integrate artificial intelligence (ML/DL) for solving clinical problems [51]. When properly trained and validated, such methods can exceed human performance and therefore lead to better patient outcomes [52]. Moreover, AI techniques can aid clinicians and improve their performance, while decreasing time spent working [53]. In the setting of breast cancer lymph node prediction, we investigated three main classification modalities: (1) radiomics, (2) clinicopathological data, and (3) a combined approach. Somewhat modest results were achieved when only clinicopathological data were used (AUC 0.74–0.77), but such an approach is more ready to be implemented in the everyday clinical workflow, because the features used are usually obtained during regular breast cancer patient workup.

The radiomics approach showed great promise, especially when dealing with MRI image data, with AUC up to 0.996 and ultrasound data with AUC up to 0.966. Alas, a common limitation of those studies were small sample sizes (Table 1 and Table 2) and questionable validation methods (no external validation cohort, cross-validation or bootstrapping) [16,17,34,35]. Additional focus must be on preventing data leakage since often multiple images are obtained from the same patient. Therefore, model performances can be overly optimistic, and can underperform when evaluated on unseen real-world data. That can be seen in the results reported by Jiang et al., where the XGBoost AUC on the test set was 0.81, which dropped to 0.70 when evaluated on an external validation cohort [47]. Moreover, one systematic review has shown that in DL radiology applications, validation scores will demonstrate a modest to substantial decrease when compared to internal performance [54]. Therefore, a general trend is a tendency to overfit the models to internal data and to “optimize for publication” [55]. To alleviate these pitfalls, more rigorous internal and external validation procedures should be implemented.

Additionally, performing multi-centric studies and increasing sample size should also provide more robust models. Several guidelines that tackle reporting in medical prediction studies have been previously published [56,57,58,59]. They demonstrate that research should report on how representative the study population is, how to properly separate training and test data (avoiding data leakage), and how to choose the proper evaluation metrics. Nevertheless, the aforementioned studies, and studies implementing AI in medicine in general, seldom utilize these guidelines.

Finally, a multi-modal approach has been presented where we combined data from multiple sources (clinicopathological and radiomics data) to obtain better performing and more robust models [9]. Combining different data modalities usually leads to better performance. Therefore, such an approach should be more extensively researched in future studies to provide optimal models for breast cancer lymph node classification [9,50].

## 8. Conclusions

To conclude, before ML/DL models can be implemented in everyday clinical practice for breast cancer lymph node classification, we need external, prospective model validation along with standardized target metric reporting. Furthermore, multicentric studies with larger sample sizes and multi-modal approaches that utilize data from different sources are also required to produce ML/DL systems that are ready for clinical use.

## Figures and Tables

**Table 2 cancers-15-02400-t002:** Summary of studies using ultrasound radiomics for breast cancer lymph node classification.

Study	Sample Size	Model	Metric
Lee et al. [34]	153	Multiple models (DenseNet, XGBoost, SVM, LR)	AUC = 0.805
Ozaki et al. [35]	300 images LN(−), 328 images LN(+)	CNN with Xception architecture	AUC = 0.966
Sun Q. et al. [36]	169	CNN de novo	AUC = 0.72 (SD ± 0.08)
Sun S. et al. [37]	479	CNNs and random forests	AUC = 0.912 [0.834, 99.0]
Tahmasebi et al. [38]	317 nodes	Google Cloud AutoML Vision	Sensitivity = 74.0%, accuracy = 69.5%
Zhang et al. [39]	952	10 ML models (XGBoost best performer)	AUC = 0.916
Guo et al. [40]	937	CNN de novo	SLNm and NSLNm sensitivity = 98.4% [96.6–100]) and 98.4% [95.6–99.9]

AUC—area under the ROC curve, CNN—convolutional neural network, SVM—support vector machine, LR—logistic regression, SLN—sentinel lymph node, NSLN—non sentinel lymph node, ML—machine learning.

**Table 3 cancers-15-02400-t003:** Summary of studies using other imaging modalities for breast cancer lymph node classification.

Study	Sample Size	Imaging	Model	Metric
Abel et al. [41]	75	Mammography	dCNN	Accuracy = 95.96%
Liu et al. [42]	401	CECT	DA-VGG19	AUC = 0.805
Yang et al. [43]	348	CECT	Deep learning signature	AUC = 0.817 [0.751–0.884]
Song et al. [44]	100	PET-CT	XGBoost	AUC = 0.890 [0.700–0.979]
Morawitz et al. [45]	303	PET/MRI	Random forest	Accuracy = 89.3%

AUC—area under the ROC curve, dCNN—deep convolutional neural network, DA-VGG19—deformable attention visual geometry group 19 neural network, CECT—contrast enhanced computer tomography, PET—positive emission tomography, MRI—magnetic resonance imaging.

**Table 4 cancers-15-02400-t004:** Summary of studies using clinicopathological features for breast cancer lymph node classification.

Study	Sample Size	Model	Metric
Dihge et al. [25]	800	Neural network	AUC = 0.74 [0.72–0.76]
Vrdoljak et al. [10]	8381	XGBoost	AUC = 0.762 [0.726–0.795)
Jiang et al. [46]	142	XGBoost	Test AUC 0.81 [0.799–0.826], validation AUC 0.700 [0.683–0.716]
Takada et al. [45]	467	ADTree	AUC = 0.77 [0.69–0.86]

AUC—area under the ROC curve.

**Table 5 cancers-15-02400-t005:** Summary of studies combining radiomics and clinicopathological features for breast cancer lymph node classification.

Study	Sample Size	Data Types	Model	Metric
Zheng et al. [9]	1342	Ultrasound and clinicopathological data	Neural network	AUC = 0.936 [0.910, 0.962]
Cheng et al. [48]	290	18F-fluorodeoxyglucose Mammi-PET, ultrasound, physical examination, Lymph-PET, and clinical characteristics	Lasso regression + Nomogram	AUC = 0.93 [0.88–0.99]
Chen et al. [49]	111	DCE-MRI and transcriptomic data	Logistic regression	AUC = 0.82

AUC—area under the ROC curve, DCE-MRI—Dynamic contrast-enhanced Magnetic resonance imaging, PET—Positive emission tomography.

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
