# Peer review of "The Role of AI in Breast Cancer Lymph Node Classification: A Comprehensive Review"

_cancers, 2023, doi:10.3390/cancers15082400_

Round 1
Reviewer 1 Report
Vrdoljak et al presented a review paper focussing on AI in breast cancer lymph node classification. This topic is very interesting to readers as it has highlighted the application of AI which is the current research everywhere. This review described AI models used for breast cancer lymph node metastasis identification, validation and the drawbacks of AI. Moreover, the review highlighted future perspectives and best practices to reach clinical setting. The review can be published after revising minor corrections as mentioned below:-
1. plagiarised text from lines 98 to 100 needs to be modified.
2. Table -1: include the name of the software used in each study.
Author Response
Dear reviewer,
Thank you for your suggestions;
1) Concerning the text in lines 98-100, this was mistakenly left behind from the mdpi template. We have removed it. Thank you for noticing it.
2) Concerning the software used, all studies are using python and the respected libraries needed for coding deep learning architectures. We have added that in the table subtext.
Reviewer 2 Report
General Comments
This is an excellent review of the field and I have no hesitation in recommending that it be published in Cancers. It is a well-written, comprehensive and credible review of AI in the classification of lymph nodes in breast cancer. The authors have not only published important work in this area but they have also done a fine job in identifying, summarising and commenting on the work of others.
Specific Comments
line 39 ... and presents future directions and ...
line 41 under Keywords, I suggest adding "AI" or "Artificial Intelligence"
line 126 ... can experiment with model architectures and ...
line 127 ... that are better optimised for the task ...
line 146 ... good performance in breast cancer ...
line 153 ... and getting overly optimistic results.
line 154 ... first approach researchers can take ...
line 157 ... different non-overlapping indexes.
line 161 ... the model's true performance.
line 166 ... original training or test data.
line 195 ... also reported excellent results, with ...
line 226 ... the sample size was small ...
line 228 A similar approach with ...
line 230 A total of 155 patients was ...
line 241 ... from MR images [18].
line 242 ... lymph nodes was used for ...
line 246 The tested 100 different ...
line 251 Table 1. The references should be in square brackets [ ] to match the text. This comment applies to Tables 2, 3, 4 and 5.
line 289 Hence, this study showed how an ...
line 352 ... and accuracy, and hence the model ...
line 393 ... on the training set and AUC of ...
line 398 ... trained on clinicopathological features ...
line 406 ... current performance of models ...
line 408 ... could improve performance.
line 412 ... they do not add additional operational costs.
line 421 ... Such an approach was ...
line 423 A model trained on both feature ...
line 426 ... also exhibited excellent model performance when trained ...
line 437 ... and genomics-only models
line 446 ... lead to the most accurate ...
line 447 ... such models is a resource intensive ...
line 454 ... an exponential growth of methods that ...
line 481 ... training and test data ...
line 487 ... performance. Therefore, such an approach should be more extensively researched in future. studies ...
References
I am concerned that many of these are incomplete in terms of the details provided. This is clearly the case for references [11], [22] and [24], and may also be the case for others -- such as [5], [8], [10], [12], [21], [29], [30], [32], [33], [36], [37], [39], [40], [41], [42], [47], [49], [50] and [54]. I would urge the authors to ensure that references include the full page ranges and be cited in a way the enables a reader to track down the reference without great difficulty.
Author Response
Dear reviewer,
Thank you for your comments and suggestions.
We have implemented the needed corrections.
Concerning the references, we have added the paginations and other information where it was missing as suggested.
Reviewer 3 Report
Interesting comprehensive review. The title, abstract,search strategy , results and discussion are synthesized appropriately. I would prefer in the search strategy section to mention in brief the number of the studies retrieved and selected.
Author Response
Dear reviewer,
Thank you for your comment.
We have added the following to search methodology:
"After performing the search, from initial 431 studies, 52 were selected for further examination. "